# Study of Dyeing Process of Hemp/Cotton Fabrics by Using Natural Dyes Obtained from *Rubia tinctorum* L. and *Calendula officialis*

**DOI:** 10.3390/polym14214508

**Published:** 2022-10-25

**Authors:** Gabriela Mijas, Mariona Josa, Diana Cayuela, Marta Riba-Moliner

**Affiliations:** 1Terrassa Institute of Textile Research and Industrial Cooperation (INTEXTER), Universitat Politècnica de Catalunya (UPC), 08222 Terrassa, Spain; 2Department of Materials Science and Engineering (CEM), Universitat Politècnica de Catalunya (UPC-ESEIAAT), 08222 Terrassa, Spain

**Keywords:** natural dyes, hemp, cotton, calendula, common madder

## Abstract

The objective of this work was to assess the possibility of dyeing a substrate composed of non-textile industrial hemp using natural dyes from common madder (*Rubia Tinctorum* L.) and calendula (*Calendula Officialis*) and tannin and alum as mordants. The substrate used for the dyeing had a 25/75 hemp/cotton composition. The hemp raw material is an agricultural by-product that was subjected to mechanical and chemical treatments in order to cottonize the fibers, blend them with cotton, and thus obtain first 40-tex open-end yarns and then a knitted fabric. The latter was subjected to different dyeing conditions by varying the dye, mordant, and method for its application, type of water, and rinsing after dyeing. Measurements of the difference (ΔE) and intensity (K/S) of color and fastness to washing and rubbing were carried out. The results showed that dyeing of a non-textile residual hemp substrate is possible, and that calendula is a good option for dyeing it with tap water, tannin-alum set in a meta-mordanting process, and rinsing after 24 h. In this way, a contribution has been made to the circular economy of the textile industry through the use of more sustainable sources and products.

## 1. Introduction

For several years, synthetic dyes from petroleum-based raw materials have predominated the textile industry due to their wide variety of colors, fastness, and low cost. Nevertheless, their production implies the consumption of high quantities of resources and energy [1]. For this reason, there is an increasing interest to adapt or develop processes with natural dyes to obtain similar results to those of synthetic dyes. In this sense, the market of natural dyes is estimated to grow with a compound annual growth rate (CAGR) of 11% during the period 2018–2024 [2].

On the other hand, cotton has been the most used worldwide natural fiber due to the comfortability and breathability of the resulting fabrics. Nevertheless, the main drawback of its use is that, currently, the availability of land for its cultivation is quite limited and it is insufficient to cover the world market demand [3], which has an incremental tendency due to the growing sensitivity of customers on sustainable goods. Some other drawbacks of cotton are the high amount of water consumed during cultivation, the necessity of using pesticides and the proper conditions for harvest (can only be grown in temperate-hot climates) [4,5]. For these reasons, the use of alternative natural fibers such as hemp can be blended with cotton to achieve similar or even better properties as pristine cotton in the final product will contribute to an increase in the sustainability of the textile process [6,7].

Conventionally, hemp fibers have been extracted in a long fiber (50–60 cm) form, with a significant quantity of short fibers. Nevertheless, in the mechanical processing of hemp straw, especially during scutching and hackling, there is a removal of fibers that are not suitable for traditional long fiber spinning due to their length but could still be properly utilized by another spinning method [8,9,10,11]. In addition, according to a study conducted by the European Industrial Hemp Association (EIHA) in 2013 [12], from the total fibers produced from non-textile industrial hemp, only 0.1% were used for clothing. This shows that there is a large amount of textile fibers that could be obtained from by-products that, subjected to the appropriate physical and/or chemical treatments and spinning methods for short fibers, could have textile applications [12]. Cottonization, as a possibility of transforming technical fibers into finer aggregates of elementary fibers, allows hemp its production and use “as-cotton” [13,14], although this process presents more difficulty than in the case of flax due to the lignin content of the hemp fibers that have been harvested after flowering and are not of a textile cultivar [15,16]. 

Dyeing is one of the most polluting stages of the entire textile process [17], and the replacement of synthetic dyes by dyes of natural origin is therefore considered to be of great interest. Natural dyes can be extracted from different biodegradable and renewable sources such as plants, animals, minerals, fungi, algae, microbes, and insects [9,10,11,12,13,14,15,16,17,18,19,20,21,22,23]. All of these dyes also have antioxidant, antibacterial, anti-allergic, antiviral, antifungal and anti-UV properties due to the presence of different functional groups present in their structure [20,24]. In addition, they are environmentally friendly, non-toxic, non-poisonous, non-carcinogenic, and non-hazardous in nature [25,26].

Common madder (*R. tinctorum* L.) grows in the west, south, and southeast parts of Europe, Africa, and South America. It has been used for dyeing textiles since 2000 B.C. [27,28]. Pigments can be extracted from the roots of common madder. Their main dye components are anthraquinones with alizarin (1,2 dihydroxy anthraquinone), the hydrolysis product of ruberythric acid. These pigments produce useful colors that have distinctive heat and light resistant properties. The anthraquinones probably contribute to the resistance of the plant against fungi in the soil [28]. Moreover, Rubia tinctorum has antimicrobial activity against some Gram (+) and Gram (−) bacteria, yeasts, filamentous fungi, and actinomycetes [29].

Pot marigold, common marigold, garden marigold, English marigold, or Scottish marigold (*Calendula officinalis*) is native to Asia and southern Europe. It has been used due to its large number of helpful properties such as antibacterial, antifungal, antiviral, anti-inflammatory, wound healing, etc. [30]. The coloring matter is in its bright orange or yellow flowers that contain lutein (a carotenoid pigment), which includes alpha- and beta-cryptoxanthin and hydroxyl groups [18,31,32].

Typically, dyes obtained from plants have no affinity for cellulose, so they need a mordant to permanently fix to the textile fibers [33] (Figure 1), resulting in improved color and color fastness [34,35]. Consequently, environmentally friendly mordants such as tannins and alum are necessary to make the dyeing process more sustainable and eco-friendlier [36,37].

On one hand, tannin is a mordant found in some plants that also produce dyes such as *Acacia nilotica* and pomegranate peel (*Punica granatum*) [25,38]. Tannins are important in the textile area because they produce mordants that are indispensable for dyeing vegetable fibers such as cotton and linen. In addition, they are very often associated in plants with yellow, orange, red, and violet, whose color is reinforced in the dye bath by their own pigment. Textile materials dyed with tannins have good washing and light fastness [39].

On the other hand, alum (KAl(SO_4_)_2_⋅12H_2_O) is the most common mordant and it has been used as a mordant for the textile dyeing of yarn, cloth, and leather in North America, England, China, Libya, Russia, and Turkey since ancient times [40]. Alum does not affect color and is in the category of brightening mordants because it tends to brighten colors. It will make yellows bright yellow and is a must for bright reds when dyed with madder [39,41].

The objective of this work was to evaluate the possibility of dyeing a fabric containing hemp of non-textile origin by using natural dyes from common madder and calendula and to study the influence of different parameters for the dyeing process such as water hardness and the mordanting process to achieve high color intensity and fastness to washing and rubbing.

## 2. Materials and Methods

### 2.1. Materials

The raw material used was hemp (*Cannabis sativa* L.) purchased from CELESA S.A. (Tortosa, Spain) and cotton (micronair 4.4) obtained from Algosur (Lebrija, Spain). Both the dyes and mordants used in this study were purchased from Lana y Telar (Sevilla, Spain). Dyes were obtained from dehydrated flowers of calendula *(Calendula Officialis)* and common madder powder (*Rubia Tinctorum*). As mordant agents, fine tannin powder (C_76_H_52_O_46_) and potassium alum crystal (KAl(SO_4_)_2_·12H_2_O) were chosen. Sodium hydroxide (NaOH) extra pure and glacial acetic acid were supplied by Scharlab (Sentmenat, Spain), hydrogen peroxide (H_2_O_2_) 50% w/v was purchased from VWR chemicals, and sodium carbonate powder ≥99.5% from Merck.

### 2.2. Preparation of Hemp/Cotton Fabric

Pre-treatment

Previously, hemp fibers were opened, washed with a non-ionic surfactant, Hostapal UH liq (Clariant, Sant Joan Despí, Spain), dried, treated in an alkaline bath mainly with NaOH 10 g/L, and then with an oxidant treatment mainly with H_2_O_2_ (50% *w*/*v*) 1 g/L. Laboratory dyeing and the fastness testing machine Linitest^®^ (Original Hanau-Heraeus) was used in these chemical processes. Once the treatments were completed, hemp was cooled, rinsed, neutralized with acetic acid 0.6 g/L, rinsed with distilled water, and dried at room temperature. When the fibers were completely dry, they were passed through a laboratory opener.

Yarn spinning

Before yarn spinning, a sizing treatment based on hemp seed oil was employed on the open treated hemp fibers to reduce the friction coefficient. First, slivers of 5 g of hemp/cotton (HE/CO) at a 25/75 weight ratio were prepared using a micro dust and trash analyzer (MDTA 3Uster, Uster, Stwitzerland). The blend was introduced three times into the MDTA conveyor belt to obtain a 1 m long homogeneous sliver.

To obtain the yarns, an Open-End rotor machine (Spintester OE rotor box SE10Slchafhorst, Fellbach, Germany) was used for spinning the slivers previously obtained. The machine was set to obtain yarns with a count of 40 tex.

Knitting

A circular knitting machine (gauge = E23, diameter = 90 cm, number of needles = 65) was used to obtain knitted fabrics from yarns. The following parameters were set: feed rate = 450 ± 10 mm/turn, length of absorbed yarn (LFA) = 0.690 cm/stitch, cover factor = 9.

### 2.3. Wet Processing

Once the knitted fabrics were obtained, different processes were applied. The aim of this stage was to determine the most suitable dyeing conditions for the fabric used. The variables studied were the effect of the mordant agent before, during, and after dyeing; water hardness; type of dye, and effect of a subsequent rinse. The procedure followed to determine the best conditions is shown in Figure 2, and Table 1 shows the nomenclature used for the samples according to the dyeing process applied in each of them.

Wet processing began with the washing of the fabrics in order to remove the sizing oils. Subsequently, the variables of the mordanting were studied, establishing the most suitable conditions for the fabrics used. Laboratory dyeing and the fastness testing machine Linitest^®^ (Original Hanau-Heraeus) were used in the wet processes. Then, the procedure for the addition of the mordant was established, whether it was before, during, or after the dyeing. Four samples were then processed and only two were rinsed after dyeing. Finally, considering the conditions established in the previous steps, the dye that contributed a greater intensity of color and the best wash and rubbing fastness to the fabrics of HE/CO 25/75 was determined.

Washing of fabrics

Before dyeing, the fabrics were washed to remove any oiling or pollutant that may have been acquired during the knitting operation. A total of 1 g/L of washing agent (Hostapal UH liq) was applied at 37–40 °C for 30 min with a liquor ratio (LR) of 1:30. Subsequently, four rinses with distilled water were carried out. The first was done at 37–40 °C and the three remaining rinses at room temperature.

Mordanting

To ensure the fixation of the dye to the fiber, all samples were subjected to two mordantings with alum. In the case of the samples in which the influence of tannin on the dyeing was studied, a mordanting with tannin was carried out prior to the alum applications. Additionally, the effect of the salts and other ions present in tap water (hardness: 355 mg/L CaCO_3_ [42]) was analyzed, so the mordanting was applied using tap water or distilled water. The nomenclature of the samples according to the process applied can be observed in Table 1.

Mordanting with tannin

A total of 300 mL of a solution of 8% tannin over weight fiber (% o. w. f.) with 5 g of fabric were placed in the Linitest vessels at 50–60 °C for 1 h. Once the fabrics were removed, two mordantings with alum were applied directly.

Mordant with alum

For the first mordanting, 5 g of the fabric and 300 mL of a solution of alum 15% o. w. f. and sodium carbonate 2% o. w. f. was applied for 1 h at 50–60 °C. For the second mordanting, a solution of alum 10% o. w. f. and sodium carbonate 2% o. w. f. was used under the same conditions. This reagent, in addition to its detergent properties, was used together with alum in order to regulate the pH and improve the absorption of the dye into the fiber. At the end of the process, the fabrics were dried at room temperature.

Dyeing process

Temperatures chosen for these processes were based on the literature about natural dyes [43,44].

o Common madder

A solution of 8% o. w. f. of the powder extracted from the common madder plant was used for dyeing 5 g of fabric at a LR of 1:60. This process was carried out for 1 h at 60 °C. The dyed fabrics were removed from the vessels and left to dry with or without further rinsing, depending on the case.

o Calendula

First, the dye was extracted from the dehydrated flower of calendula. A solution containing calendula flower 40 g/L was prepared and boiled in a flask with a reflux condenser and magnetic stirring for 30 min. Afterward, 300 mL of the filtered dye solution and the fabrics were introduced into the Linitest vessels at LR of 1:60 for 1 h at 100 °C. Finally, the fabrics were removed and left to dry with or without further rinsing, depending on the case.

### 2.4. Characterization of Dyed Samples

Colorimetry

These determinations were conducted on dye substrates using a reflectance spectrophotometer CM-3600D (Minolta, Madrid, Spain) with a measurement geometry of d/10 (diffuse illumination/angle of view 10°), illuminant D65, observer 10°, large measurement area including 100% of the ultraviolet radiation. The results obtained were the average of four measurements on the same sample.

Dye Fixation

The color intensity acquired was determined by reflectance measurements to obtain the K/S value from the formula developed by Kubelka and Munk (Equation (1)).
(1)$$f\left(R\right)=K/S=\frac{{\left(1\;-R\right)}^{2}}{2R}$$
where K is the absorption coefficient; S is the scattering coefficient; and R is the fabric reflectance at λ_max.

K/S value is calculated with reference to the sample without dye, so the corrected value is given by Equation (2), where subscripts *d* and *u* refer to the dyed sample and undyed sample, respectively.


(2)
$$K/S={\left(K/S\right)}_{d}-\;{\left(K/S\right)}_{u}={\left(\frac{{\left(1\;-R\right)}^{2}}{2R}\right)}_{d}-\;{\left(\frac{{\left(1\;-R\right)}^{2}}{2R}\right)}_{u}$$


Color quantification

Colorimetric analysis and the calculation of the color coordinates of the CIELab 1976 color space including the correlations of brightness, chroma, and hue were carried out according to the UNE-EN ISO 105-J01:2000 “Tests for color fastness. Part J01: General principles for measuring of surface color” [45].

Color difference

Color difference was calculated according to UNE-EN ISO 105-J03:2009. “Color fastness testing. Part J03. Calculation of color differences” [46]. 

UV–Vis spectroscopy

The UV–VIS spectroscopy technique was used as a comparative tool between the different samples to determine the dyeing yield by analyzing the residual dyeing baths. A spectrophotometer UV-1800 (Shimadzu, Kioto, Japan) was used, and the cells used for the test were made of quartz and had an optical path length of 10 mm.

Fastness tests

In all of the fastnesses test performed, a grey scale was used to assess the change in color according to the UNE-EN ISO 105-A02 standard [47] and to assess the staining according to the UNE-EN ISO 105-A03 standard [48].

For washing fastness, the method was that indicated in the standard “UNE-EN ISO 105-C06:2010: Color fastness to domestic and commercial laundering” [49] under the A1S test conditions shown in Table 2. A color fastness tester Rotowash^®^ (SDL Atlas, Rock Hill, United States of America) and the control sample fabrics indicated in Table 3 were used.

For rubbing fastness, the procedure followed was that detailed in the standard “UNE-EN ISO 105-X12:2016: Tests for color fastness—Part X12: Color fastness to rubbing” [50]. Tests were performed with the samples wet and dried.

## 3. Results and Discussion

### 3.1. Preparation of HE/CO Fabric

It is worth mentioning that the study was carried out with non-textile industrial hemp waste, which is used primarily to obtain seeds and oils and has been used in the paper industry. The reuse of this material provides an alternative as a raw material for the textile industry within the principles of the circular economy. However, raw hemp incorporated with relevant amounts of woody wastes makes it difficult to treat and obtain a spinnable material. With this knowledge, finding the adequate conditions to obtain yarns will then make them easily extrapolated to a textile degree hemp.

With the aim to prepare knitted fabrics, the steps depicted in Figure 3 were followed, starting with a chemical treatment, and finishing with the knitting machine.

### 3.2. Study of Mordanting Variables

Although it is common industrially for mordanting with two products to be carried out in a single process, in this laboratory study, each mordanting process was carried out separately because the aim was to evaluate the behavior of the substrate with each mordant without interference, considering that the hemp fibers that make up the fabric to be dyed did not come from a textile variety.

Due to the low fastness of the natural dyes, a pre-mordanting process with tannin and alum and only with alum was considered and studied. For this, samples containing tannin in the mordanting had a higher color difference with respect to the untreated fabric (Figure 4a and Table 4). Tannin is a natural component that contains chromophore groups that contribute to the increase in the colorization of the samples. It should be noted that the fabric treated with distilled water and tannin had a lower color difference than those that had been treated with tap water. The electrolytes present in tap water may foster an ionic motion of the elements of the reactions, so in this case, it could have helped to obtain a better absorption of the mordant. In the case of the fabrics without tannin, the difference in color between the samples treated with tap and distilled water was lower, in concordance with the observed results.

When the color intensity was analyzed, samples without tannin but with alum kept their initial color or even adopted a higher white degree and showed K/S values (Figure 4b) even lower than the original sample. These changes could be attributed to the addition of Na_2_CO_3_, which would help to remove some impurities due to its inherent good detergent properties and its pH-regulating effect. On the other hand, samples mordanted with tannin adopted a brownish color, subsequently increasing the overall K/S value. Once the pre-mordanting was carried out, the samples were dyed.

Samples dyed with common madder are shown in Figure 5a. Considering the undyed sample, the samples that did not contain tannin had a higher color difference with respect to the others (Table 5). Samples treated with hard water showed a greater difference, although minimal.

In this context, samples with tannin had higher K/S values (Figure 5b) compared to the samples treated only with Na_2_CO_3_ and alum, indicating a greater dye absorption, as previously observed.

Color permanency of the dyed fabrics was determined by rubbing and washing fastness tests (Table 6). In general, all fabrics had a high change in color. This could be explained by the fact that at the end of the dyeing process, there could have been dye that was not absorbed by the fibers and remained on their surface, and since no subsequent washing was carried out, this was finally removed during the washing resistance test, causing a high color change. However, it could be said that the samples treated with tannin and distilled water resulted in better results. Additionally, all fabrics had a relatively low staining on cotton. Regarding the rubbing fastness in dry conditions, very similar results between the four samples were obtained. In the four cases, dry rubbing fastness results were higher than the wet homologous.

The dyed samples with calendula are shown in Figure 6a. Similar variations regarding the color differences with respect to common madder were obtained (Table 7).

K/S values corroborated that there was a small variation in the dyed fabrics depending on the type of water used (Figure 6b). Samples treated with tap water had slightly higher values than those treated with distilled water, which implied that the first ones were darker.

With regard to the washing and rubbing fastness tests, fabrics with tannin and tap water experienced a slight change in color (Table 8). Concomitantly, the sample processed with distilled water had a very high change in color. However, all of the results were similar, and all fabrics resulted in a very low staining, even null after washing. This could be due to the chemical structure of calendula, which is a rather linear and planar molecule like cellulose, which allows it to be introduced between the cellulose planes, easing its absorption into the hemp fiber. Dry rubbing fastness resulted in equivalent results between the four samples and in general, it could be said that they had good results. In wet, the best results were obtained when tannin and distilled water were used in the pre-mordanting process. In all four cases, the dry rubbing fastness was better than the wet rubbing fastness.

Analyzing the mordanting variables, in the case of calendula, the best results were found in sample 1CTD, which was submitted to a pre-mordanting with tannin and tap water. In the same sense, regarding common madder, the fabric that resulted in the more promising colorimetric results was 1RTD, which was mordanted with tannin and tap water. Regarding the best wash and rubbing fastness, the 1RTV and 1RTD samples provided minimal differences, which means that the type of water did not have a significant influence. For this reason, it was considered that the most suitable conditions for the next steps were mordanting with tannin and the use of tap water.

### 3.3. Study of the Best Conditions for Mordant Application

Mordanting treatment can be applied at different times of the dyeing process, affecting the results. For this, the influence of the moment for mordant application (tannin) was studied either before, during (meta-mordanting), or after the dyeing process using common madder and calendula as dyes.

Samples dyed with common madder by using tannin and alum as mordants are shown in Figure 7a. The obtained results show that the sample in which the mordant was applied before dyeing had a greater difference in color with respect to the untreated fabric (Table 9). With these results, it might be assumed that by applying a mordant prior to dyeing, a greater number of bonding points are created because there are no other products or compounds that may interfere with it, resulting in a higher absorption. Thus, the most influential factor for a greater absorption was tannin and not the dye.

With respect to the K/S values (Figure 7b) of the sample with the pre-mordant (1RTD), they presented a “shoulder” that corresponded to the red wavelength but already approached the yellow. The post-dyeing sample (2RTD) had a “shoulder” at 550 nm corresponding to the blue lilac, and at 500 nm, it adopted a color more similar to the burgundy. Finally, the sample with mordanting during dyeing (3RTD) had two small “shoulders” and another more pronounced, resulting in a red-brown color. Additionally, it is worth mentioning that the fabric with a previous mordanting had a higher K/S than the fabric with post-mordanting, indicating that the operation of mordanting after the dyeing process resulted in a lower capacity of color absorption.

Furthermore, the sample with mordanting during the dyeing process showed higher values regarding the washing and rubbing fastness than the rest of the samples (Table 10). All of them had low or null staining on the cotton and wool standard substrates. Focusing on the rubbing tests, higher values of fastness were obtained in both dry and wet, with the operation of mordanting during the dyeing process.

Samples that were dyed with calendula by using tannin as the mordant are shown in Figure 8a. In this case, the sample with the greatest color difference with respect to the undyed one (Table 11) and the highest K/S (Figure 8b) values was the fabric with mordanting applied at the same time as dyeing. Presumably, one fact that could affect the results was the temperature of the process because the mordanting was carried out at the same temperature of the dyeing process, being 100 °C, in contrast to the other two cases where the pre- and post-mordanting were undertaken at 60 °C.

The aforementioned results agreed with the washing and rubbing fastnesses (Table 12). Although in general, in the case of the rubbing fastness, higher values when dry rather than wet were obtained.

As a summary of this section, in the case of common madder, the combination of conditions that allowed us to obtain a fabric with the highest color difference, the highest K/S values, and therefore the highest dye intensity was sample 1RTD with mordanting before dyeing. However, this sample resulted in the worst washing and rubbing fastness, which meant that there was a high absorption of dye but that most of it was not fixed. Otherwise, the fabric with the highest values of fastness was the 3RTD with a mordanting operation during dyeing process. Moreover, it could be concluded that by applying the mordant during dyeing, the fabric absorbed as much tannin as in the case of the pre-mordanting. The fact that it absorbed a greater quantity of tannin could cause a perception of a greater intensity of color.

In the case of calendula, the fabric with the highest color intensity and the more promising values of wash fastness but the lowest of rubbing was 3CTD, with a mordanting during dyeing.

In conclusion, fabrics with meta-mordanting were those that resulted in more promising results.

### 3.4. Study of the Need for Rinsing after Dyeing

The influence of a rinse after dyeing was also studied. Samples dyed with common madder with and without a later rinsing are shown in Figure 9a. When samples were compared to the untreated sample, the sample with that was rinsed was the one with the lowest color difference, presumably indicating color fading with respect to the not rinsed one and, consequently, revealing that there was adsorbed and unfixed dye on the textile (Table 13). This observation agreed with the K/S values (Figure 9b), where a vanishing of the color was translated into a decrease in the intensity of the signal.

The residual dyeing bath had a greater absorbance than the rinsing bath (Figure 9c), indicating that the first of them contained a higher amount of dye. The two curves had small shoulders, showing an absorption in the blue and green region and therefore, the appreciated color was red.

Afterward, the washing and rubbing tests were in concordance with the observed results, showing a slight increase in their values when the sample was rinsed due to the previous elimination of the adsorbed dye (Table 14).

Similarly to common madder, in the case of calendula, the color faded away slightly after rinsing (Figure 10a and Table 15). The rinsing operation removes the adsorbed and unfixed dye in the fabrics. 

K/S values were significantly higher for the sample without rinsing, showing a small shoulder corresponding to near the yellow wavelengths, however, the other sample did not indicate a yellowish turn in the color of the sample (Figure 10b). In the case of the absorbance of baths, the residual dyeing bath had higher absorbance than the rinsing bath, and significant peaks were not observed but there were strong absorptions between 200 and 400 nm in both cases (Figure 10c). This was because the d calendula dye provided a yellowish coloration.

Regarding the washing and rubbing fastness tests, a slight increase could be observed in comparation with common madder (Table 16).

In summary, in the case of dyeing with common madder and calendula, the fact of not applying a rinse just after finishing the dyeing process allowed for a better absorption of the dye on the fiber. Therefore, the best results were 3RTD and 3CTD, that is, without rinsing.

## 4. Conclusions

Through the analysis of results, it has been possible to show that a fabric containing a residual non-textile industrial hemp is susceptible to be dyed using natural dyes. In this way, a more sustainable process has been established that contributes to the circular economy of the textile chain.

The mordant is an important variable for this type of dyeing and the results showed that the tannin–alum combination was effective. The type of water used in the mordanting process also influenced the results, with better values when tap water was used. Meta-mordanting proved to be the most appropriate for these types of dyes and mordants. When mordanting was carried out simultaneously with dyeing, the number of reactions between the different components of the bath with the substrate was greater, favoring the absorption of the dye by the cellulosic fibers.

Common madder behaved differently from calendula as the latter dye has a rather linear and planar structure that facilitates its bonding with the cellulosic fibers, unlike common madder, which has a more branched structure.

The fabric dyed with common madder had a color intensity in the red region (600 nm), in the blue wavelength (450 nm), and slightly more in the yellow (525 nm). Meanwhile, the fabric with calendula had more color intensity in the red wavelength, although below the common madder curve. In addition, at the yellow wavelength, it had a lower K/S value compared to common madder.

When comparing the effect of each of the two dyes on the fabric with respect to the untreated fabric, the common madder dyed fabric had a slightly higher color difference than the calendula dyed fabric. However, it can be concluded that calendula is a good choice for dyeing this type of hemp/cotton fabric since it presented better results in terms of washing and rubbing fastness. It is recommended to wait at least 24 h before rinsing.

## Figures and Tables

**Figure 1 polymers-14-04508-f001:**
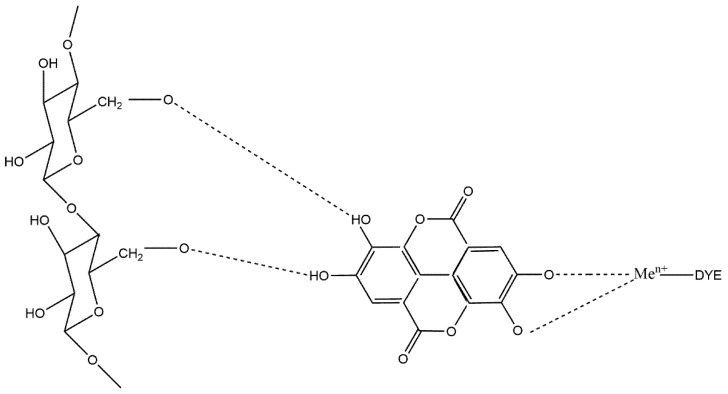
Mechanism of cellulose with tannin, metallic mordants, and dye.

**Figure 2 polymers-14-04508-f002:**
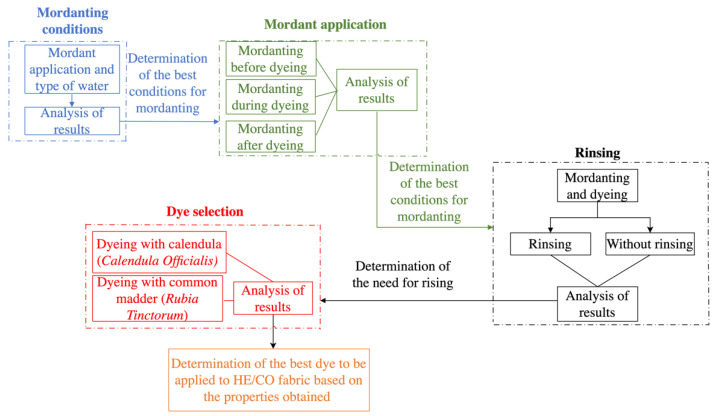
Procedure for the application of natural dyes on hemp/cotton fabrics.

**Figure 3 polymers-14-04508-f003:**
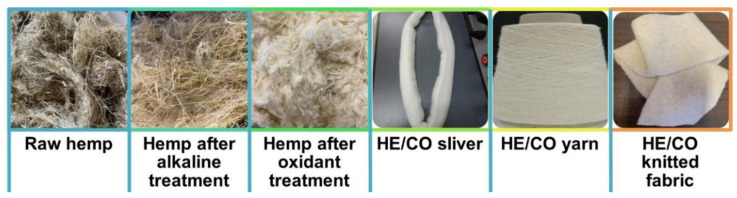
Steps followed to obtain the HE/CO 25/75 knitted fabrics.

**Figure 4 polymers-14-04508-f004:**
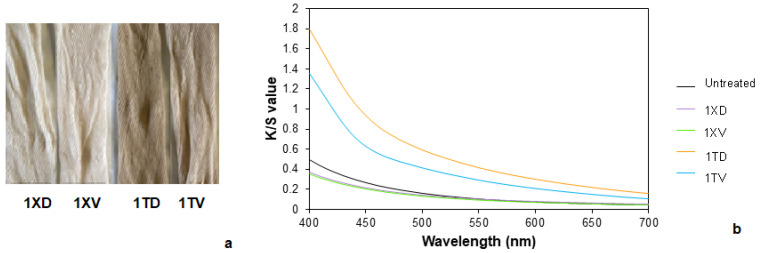
(**a**) Pre-mordanted samples and (**b**) the K/S values of the untreated and pre-mordanted samples.

**Figure 5 polymers-14-04508-f005:**
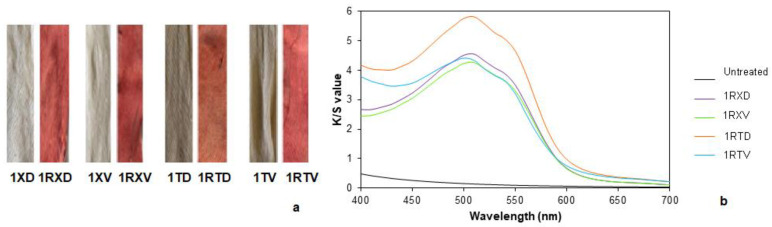
(**a**) Pre-mordanted samples before and after dyeing with common madder and (**b**) the K/S values of the pre-mordanted samples after dyeing with common madder.

**Figure 6 polymers-14-04508-f006:**
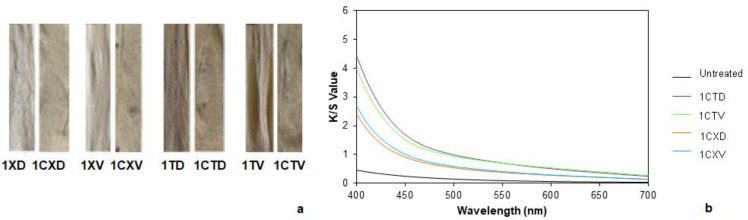
(**a**) Pre-mordanted samples after and before dyeing with calendula and (**b**) K/S values of the pre-mordanted samples after dyeing with calendula.

**Figure 7 polymers-14-04508-f007:**
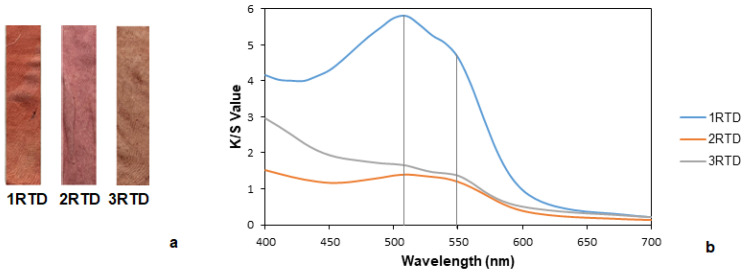
(**a**) Fabrics dyed with common madder with pre-mordanting, post-mordanting, and meta-mordanting (from left to right, respectively) and (**b**) K/S values of fabrics dyed with common madder with pre-mordanting, post-mordanting, and meta-mordanting.

**Figure 8 polymers-14-04508-f008:**
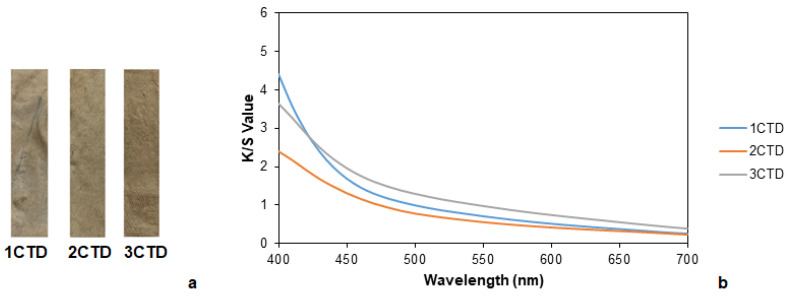
(**a**) Fabrics dyed with calendula with pre-mordanting, post-mordanting, and meta-mordanting (from left to right, respectively) and (**b**) K/S values of fabrics dyed with calendula with pre-mordanting, post-mordanting, and meta-mordanting.

**Figure 9 polymers-14-04508-f009:**
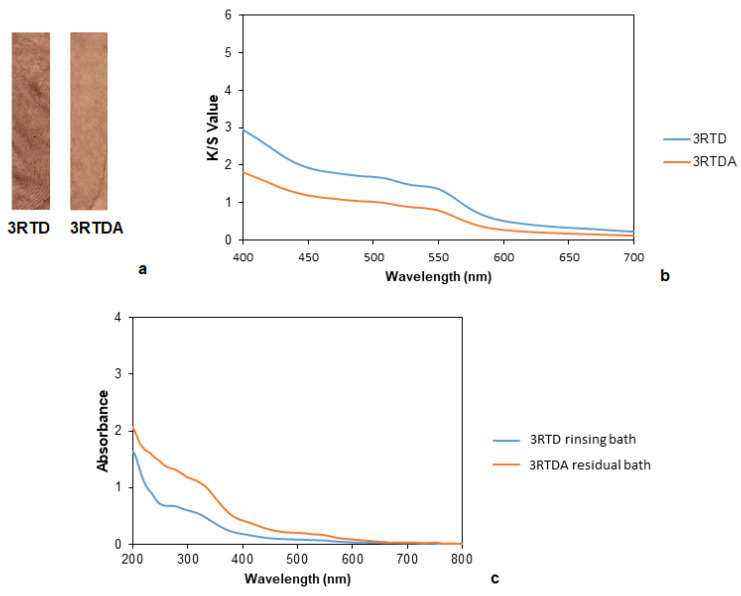
(**a**) Samples dyed with common madder without and with rinsing, (**b**) K/S values of samples dyed with common madder without (3RTD) and with rinsing (3RTDA). (**c**) Absorbance curves of the residual dyeing bath and rinsing bath of the samples dyed with common madder.

**Figure 10 polymers-14-04508-f010:**
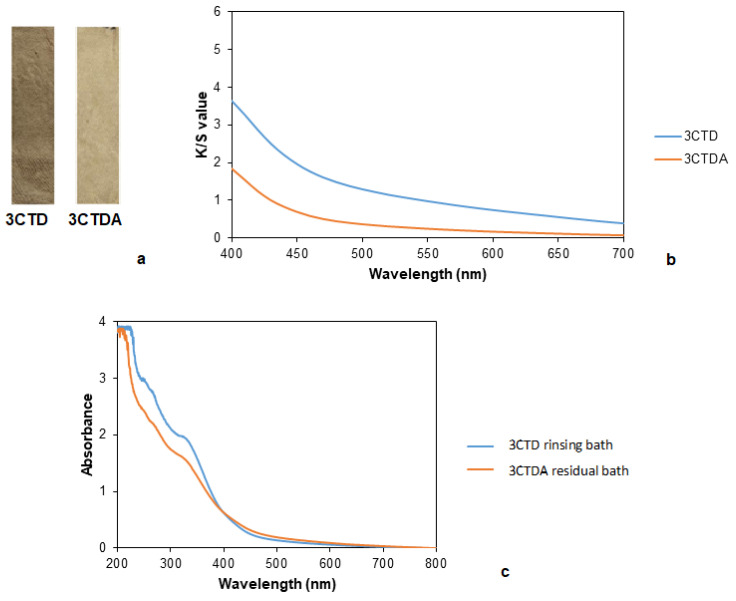
(**a**) Samples dyed with calendula without and with rinsing, (**b**) K/S values of the samples dyed with calendula without (3CTD) and with rinsing (3CTDA). (**c**) Absorbance curves of the residual dyeing bath and rinsing bath of the samples dyed with calendula.

**Table 1 polymers-14-04508-t001:** Nomenclature of the samples according to the process applied.

Sample Name	Mordanting	Dye	1 Mordanting with Tannin	2 Mordantings with Alum	Water	Rinse
Untreated	No	-	-	-	-	-
1XD	Before	-	No	Yes	Tap	-
1XV	Before	-	No	Yes	Distilled	-
1TD	Before	-	Yes	Yes	Tap	-
1TV	Before	-	Yes	Yes	Distilled	-
1RXD	Before	C. madder	No	Yes	Tap	No
1RXV	Before	C. madder	No	Yes	Distilled	No
1RTD	Before	C. madder	Yes	Yes	Tap	No
1RTV	Before	C. madder	Yes	Yes	Distilled	No
1CXD	Before	Calendula	No	Yes	Tap	No
1CXV	Before	Calendula	No	Yes	Distilled	No
1CTD	Before	Calendula	Yes	Yes	Tap	No
1CTV	Before	Calendula	Yes	Yes	Distilled	No
2RTD	After	C. madder	Yes	Yes	Tap	No
3RTD	During	C. madder	Yes	Yes	Tap	No
2CTD	After	Calendula	Yes	Yes	Tap	No
3CTD	During	Calendula	Yes	Yes	Tap	No
3RTDA	During	C. madder	Yes	Yes	Tap	Yes
3CTDA	During	Calendula	Yes	Yes	Tap	Yes

**Table 2 polymers-14-04508-t002:** The A1S test conditions according to UNE-EN ISO 105-C06:2010.

Test Number	Temperature (°C)	Volume of Bath (mL)	Active Chlorine (%)	Sodium Perborate (g/L)	Time (min)	Number of Steel Balls	pH Adjustment
A1S	40	150	No	No	30	10	unadjusted

**Table 3 polymers-14-04508-t003:** The control samples according to UNE-EN ISO 105-C06:2010 A1S.

Test number	Fiber to Test	Control Sample Fabric 1	Control Sample Fabric 2
A1S	Cotton	Cotton	Wool
	Hemp	Cotton	Wool

**Table 4 polymers-14-04508-t004:** Colorimetric values obtained from the study of mordanting variables considering the untreated sample as a reference.

Sample	Tannin	Type of Water	Color Difference (∆*E*)
Untreated	NO	-	-
1XD	NO	Tap	3.1
1XV	NO	Distilled	3.3
1TD	YES	Tap	13.9
1TV	YES	Distilled	9.5

**Table 5 polymers-14-04508-t005:** Colorimetric values obtained of the pre-mordanted samples dyed with common madder.

Sample	Tannin	Type of Water	∆*E* Respect to Pre-Mordanted Sample	∆*E* Respect to Untreated Sample
1RXD	NO	Tap	48.8	47.8
1RXV	NO	Distilled	48.1	47.0
1RTD	YES	Tap	37.6	49.7
1RTV	YES	Distilled	36.3	44.6

**Table 6 polymers-14-04508-t006:** Results of the washing and rubbing fastness tests of the pre-mordanted samples after dyeing with common madder.

Sample	Washing			Rubbing	
	Color Change	Staining		Dry	Wet
		Cotton	Wool		
1RXD	1	4–5	4–5	3	1–2
1RXV	1	4–5	4	3	2
1RTD	1	4–5	4–5	3	1
1RTV	1–2	4–5	4	3–4	2

**Table 7 polymers-14-04508-t007:** Colorimetric values obtained for the pre-mordanted samples dyed with calendula.

Sample	Tannin	Type of Water	∆*E* Respect to Pre-Mordanted Sample	∆*E* Respect to Untreated Sample
1CXD	NO	Tap	15.3	13.4
1CXV	NO	Distilled	17.0	14.7
1CTD	YES	Tap	7.6	21.4
1CTV	YES	Distilled	11.0	20.4

**Table 8 polymers-14-04508-t008:** Results of the washing and rubbing fastness tests of the pre-mordanted samples after dyeing with calendula.

Sample	Washing			Rubbing	
	Color Change	Staining		Dry	Wet
		Cotton	Wool		
1CXD	2–3	5	5	4	3–4
1CXV	2–3	5	5	4	3–4
1CTD	3–4	5	5	4–5	3
1CTV	1	5	5	4–5	4

**Table 9 polymers-14-04508-t009:** Colorimetric values obtained for the samples dyed with common madder with tannin and alum as the mordants.

Sample	Tannin	Type of Water	Mordant Application	∆*E* Respect to Untreated Sample
1RTD	Yes	Tap	Pre-dyeing	49.7
2RTD	Yes	Tap	Post-dyeing	29.6
3RTD	Yes	Tap	During-dyeing	29.2

**Table 10 polymers-14-04508-t010:** Results of washing and rubbing fastness tests of samples with pre-mordanting, post-mordanting and mordanting during dyeing with common madder.

Sample	Washing			Rubbing	
	Color Change	Staining		Dry	Wet
		Cotton	Wool		
1RTD	1	4–5	4–5	3	1
2RTD	2	5	4–5	3–4	3
3RTD	2–3	5	4–5	4	2–3

**Table 11 polymers-14-04508-t011:** Colorimetric values obtained for the samples dyed with calendula with tannin as the mordant.

Sample	Tannin	Type of Water	Mordant Application	∆*E* Respect to Untreated Sample
1CTD	Yes	Tap	Pre-dyeing	21.4
2CTD	Yes	Tap	Post-dyeing	18.0
3CTD	Yes	Tap	During-dyeing	24.6

**Table 12 polymers-14-04508-t012:** Results of the washing and rubbing fastness tests of samples with pre-mordanting, post-mordanting, and mordanting during dyeing with calendula.

Sample	Washing			Rubbing	
	Color Change	Staining		Dry	Wet
		Cotton	Wool		
1CTD	3–4	5	5	4–5	3
2CTD	4	5	5	4–5	3–4
3CTD	4–5	5	5	4	2–3

**Table 13 polymers-14-04508-t013:** Colorimetric values obtained for the samples dyed with common madder without and with rinsing.

Sample	Tannin	Type of Water	Moment for Mordant Application	Rinsing	(∆*E*) Respect to Untreated Sample
3RTD	Yes	Tap	During-dyeing	No	29.2
3RTDA	Yes	Tap	During-dyeing	Yes	22.6

**Table 14 polymers-14-04508-t014:** Results of the washing and rubbing fastness tests of the samples dyed with common madder with or without rinsing.

Sample	Washing			Rubbing	
	Color Change	Staining		Dry	Wet
		Cotton	Wool		
3RTD	2–3	5	4–5	4	2–3
3RTDA	2	5	4–5	4–5	3

**Table 15 polymers-14-04508-t015:** Colorimetric values obtained for samples dyed with calendula without and with rinsing.

Sample	Tannin	Type of Water	Moment for Mordant Application	Rinsing	(∆*E*) Respect to Untreated Sample
3CTD	Yes	hard	During-dyeing	No	24.6
3CDA	Yes	hard	During-dyeing	Yes	10.2

**Table 16 polymers-14-04508-t016:** Results of the washing and rubbing fastness tests of the samples dyed with calendula with or without rinsing.

Sample	Washing			Rubbing	
	Color Change	Staining		Dry	Wet
		Cotton	Wool		
3CTD	4–5	5	5	4	2–3
3CTDA	3	5	5	4–5	3

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
