# Peer review of "Study of Dyeing Process of Hemp/Cotton Fabrics by Using Natural Dyes Obtained from Rubia tinctorum L. and Calendula officialis"

_polymers, 2022, doi:10.3390/polym14214508_

Round 1

Reviewer 1 Report

This manuscript is entitled " Study of dyeing process of hemp/cotton fabrics by using natural dyes obtained from Rubia Tinctorum L. and Calendula Officialis". These data are interesting, but some points still need to correct before publication.

1.     Please check spelling mistakes and the English language throughout the text.

2.     Abstract: please rewrite the main results and the purpose

3.     Please check all figures and correct them properly

4.     Please check all units

5.     Introduction: The novelty and the advance added to the area must be clearly stated. Particularly Introduction could be enlarged. These things are missing.

6.     Please check all units in order to be similar

7.     Environmental viability assessment should be added.

8.     References can be added from the host journal.

9.     Please check the 3.4 section (Error! Reference source not found.).

10.  Conclusion: please add the key points with the further implication

Author Response

1. Please check spelling mistakes and the English language throughout the text.

Thanks for your suggestion. We have check the spelling mistakes and improved the English language of the text.

2. Abstract: please rewrite the main results and the purpose

The abstract has been rewritten and more emphasis has been given to the purpose and results of the work.

3. Please check all figures and correct them properly

Thanks for your suggestion. The quality of figures has been improved.

4. Please check all units

Thanks for your suggestion. The units have been checked in order to be similar

5. Introduction: The novelty and the advance added to the area must be clearly stated. Particularly Introduction could be enlarged. These things are missing.

Thanks for your suggestion. More focus has been given to the novelty of the work and some additional information has been added to the introduction.

6. Please check all units in order to be similar

Thanks for your suggestion. The units have been checked in order to be similar

7. Environmental viability assessment should be added.

Thanks for your suggestion. Although we also consider that the environmental viability assessment is an important point, this is not the main objective of the work. It is important to mention that the substrate used for the dyeing comes from an agricultural waste that has been revalued and that natural dyes have been used in order to make the process more sustainable. And considering that all the materials have been obtained from local sources, it is contributing to a circular economy by decreasing the environmental impact.

8. References can be added from the host journal.

Some references from the host journal have been included

9. Please check the 3.4 section (Error! Reference source not found.)

The error has been corrected and the missing reference has been added.

10. Conclusion: please add the key points with the further implication

Thanks for your suggestion. The conclusions have been rewritten and more emphasis has been placed on the main findings of the work.

Reviewer 2 Report

The manuscript considers the preparation of knitted fabrics from hemp/cotton yarns and analyzes the dyeing process with natural dyes from Common madder and Calendula. Different dying conditions are varied as mordanting, water quality and rinsing of dyed fabric to obtain intensive fabric colour with high fastness.  

The authors should emphasize the novelty of the work more clearly.
The tannin source and producer and the producer of the potassium alum should be stated in the experimental part.
There are many typographic errors, such as “rising” instead of “rinsing” in Figure 2 and the manuscript Conclusion; “mainly with H2O2”.
The equations from 1 to 9 can be given as supplementary data.

Author Response

1. The authors should emphasize the novelty of the work more clearly.

Thanks for you suggestion. More focus has been given to the novelty and results of the work.

2. The tannin source and producer and the producer of the potassium alum should be stated in the experimental part.

Although the request has been made, we have not yet obtained an answer on the origin of the tannin. The supplier of both tannin and alum are described in section 2.1.

3. There are many typographic errors, such as “rising” instead of “rinsing” in Figure 2 and the manuscript Conclusion; “mainly with H2O2”.

We have checked the spelling mistakes and chemical formulas through the text.

4. The equations from 1 to 9 can be given as supplementary data.

Thanks for for your suggestion. In fact, we have thought it convenient to eliminate some of them from the text because these equations are clearly explained in the UNE-ISO standards to which reference has been previously made.

Reviewer 3 Report

Dear Authors,

the manuscript is prepared according to the journal Polymers instructions. It is well structured, but some corrections and suggestions about the improvement of the manuscript are directly put in comments in the pdf version of the manuscript. 

Kind regards,

The reviewer

Author Response

Dear reviewer,

We appreciate all the comments you have made to improve the quality of our work. As you will notice in the new version of the manuscript, almost all your suggestions have been taken into consideration, but there are some points to remark.

Page 6, equation 9: Explanation of parameters L*, a* and b* is missing as well as explanation of index 1 and index 2 is missing.

We have thought it convenient to eliminate the equations from 2 to 9 from the text because these equations are clearly explained in the UNE-ISO standards to which reference has been previously made.

Page 15: The results show this, which is in contrast to the purpose of dyeing process. The dyeing process should be performed from technological, economical and ecological point of view in such manner that the highest adsorption and fixation of the dye is obtained on dyed fabric, resulted in high wet fastness properties of the fabric during use. The later is in close correlation with the functional groups on fabric, structure of the dye and the strength of dye-fibre interactions.

We agree that the purpose of the dyeing process is to obtain a product with high dye fixation and high wet fastness because it is intended that the fabric keeps its initial properties during its useful life. By no means it has been said that a rinse should not be carried out after dyeing, but it has been suggested to wait for a period of 24 h before carrying it out so that the interactions between the fiber and the dye have more time to occur. Although an oven drying could be carried out, it has not been desired to increase the energetic cost of the process.

Kind regards

The authors